# Modelling Income Drivers in Peripheral Municipalities: The Case of Italian Inner Areas

**Luca Romagnoli [1,*], Paola Di Renzo [1] and Luigi Mastronardi [2]**

1   Department of Economics, University of Molise, 86100 Campobasso, Italy
2   Department of Biosciences and Territory, University of Molise, 86090 Pesche, Italy
*   Correspondence: luca.romagnoli@unimol.it

**Abstract:** The paper provides new evidence on the relationship among per capita incomes, local expenditures and territorial economic structure in Italian inner areas. The study area comprises the municipalities belonging to three Italian regions, namely, Marche, Abruzzo and Molise. The methodology employed involves a panel multilevel regression model, in order to investigate both on territorial and time changes. The period under analysis covers 2008–2016, the years following the outbreak of the global crisis. The results highlight the importance of three public expenditure categories—Environment protection and planning, Tourism, and Cultural heritage—on the growth of per capita incomes. Regional economic structure also plays an important role, especially through the rate of employees in the industrial sector. In order to increase the effectiveness of local public policies, a re-allocation of global expenditures among its various components might be recommended. Another suggestion concerns the implementation of integrated policies oriented both to tourism and to the enhancement of territorial assets.

**Keywords:** per capita incomes; Italian inner areas; public expenditure; panel multilevel regression

## 1. Introduction

This paper proposes a study on per capita incomes and their key drivers in Italian inner areas, as classified by the "Strategia Nazionale per le Aree Interne" (SNAI, Inner Areas National Strategy) [1] in the 2008–2016 period, the years spanning the outbreak of the global financial crisis and the beginning of the economic recovery in Italy.

The crisis provides a fertile ground for research on well-being and its linkages with policies and territorial features. During the crisis, the Italian economy suffered a decrease in real gross domestic product (GDP), an increased unemployment rate, an increased poverty rate in absolute and relative terms, and augmented inequalities [2]. The combined effect of these indicators results in lower well-being levels [3,4].

Per capita incomes can be considered as a proxy of well-being. Indeed, the debate regarding the measurement of national well-being has a longstanding tradition [5–7] and is still open. Recently, the United Nations (UN) Sustainable Development Goals (SDGs), identified in the UN 2030 Agenda for Sustainable Development [8], have further shifted attention to the concept of sustainable well-being [9]. According to the Organisation for Economic Co-operation and Development [10], individual well-being can be subdivided into material living conditions and quality of life. The first concept is based on the economic situation; in this respect, per capita incomes are a fundamental component of well-being level [11]; in the short run, incomes are positively correlated to well-being [12]. Quality of life includes several factors, such as health, education, environmental quality, and others. Scholars have also dedicated a great deal of attention to the concept of subjective well-being, in which the economic component is only marginally considered [13–15]. Nevertheless, income remains a widely used variable in economic studies, particularly those regarding economic growth.

The literature has considered the determinants of income almost exclusively at the national and regional levels. Several studies regarding the convergence process [16–21] have included a large number of explanatory variables, such as public expenditures, human capital, technological innovation, population dynamics, sectoral structure/employment, and others. As to public expenditures, there is some debate on their impacts on GDP and/or income. It is important to distinguish among the components of expenditure, in particular between productive and non-productive government expenditures [22]. Productive spending has an effect on incomes [23], while non-productive expenditure fails to do so [24,25]. Human capital and education influence economic growth significantly [26,27]. Technology is a more important factor than education in convergence of per capita income levels [28]. Local economic structure is relevant for development [29–31]; in particular, the services sector is associated with positive economic performance [32,33]. Population size and growth have been found to be positively correlated to economic development [34].

In the studies concerning the determinants of economic growth and income levels, different econometric models have been employed. The spatial dimension of the data has been often taken into account, both from a static [35–37] and from a dynamic perspective [38–40]. Some models adopt a multilevel approach [41–43]. In a time-series multivariate approach, vector autoregressive (VAR) models have also been applied [44,45].

The Italian SNAI defines inner areas (IAs) based on the travel time needed for people to reach the main centres, termed Poles and Inter-municipal poles. These are the single or grouped municipalities where essential services (education, health, and mobility) are located [1]. Four other classes are identified: (1) Urban belts, that is, municipalities that are fewer than 20 minutes' distance from the nearest Pole; (2) Intermediate areas, that is, those whose distance is between 20 and 40 min; (3) Peripheral areas, that is, those that are more than 40 and 75 min away; and (4) Ultra-peripheral areas, which are more than 75 min away. Poles, Inter-municipal poles, and Urban belts are defined as Centres. The remaining three classes form the so-called Inner areas. It is important to point out that IAs' peripherality is not only of a geographical or physical kind, but also depends upon a lack of "citizenship rights" [46,47]. The distribution of Italian municipalities according to SNAI is illustrated in Figure 1.

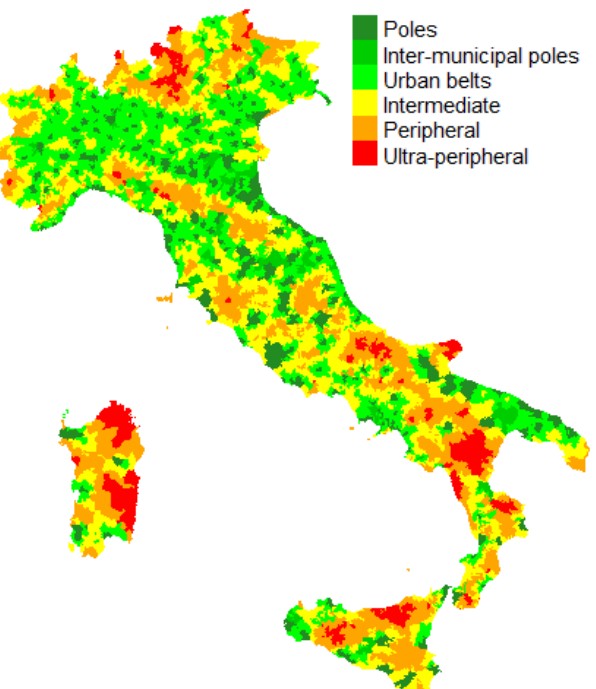

**Figure 1.** Distribution of Italian municipalities according to SNAI.

In 2014, when SNAI was implemented, IAs included 4185 municipalities, constituting 52% of total Italian municipalities. Their population amounted to more than 13 million inhabitants, around 22% of the national population, mainly residing in small villages.

IAs largely coincide with mountainous, high hills, and rural areas, and present strong criticalities from social, economic, and environmental perspectives [48].

A "place-based approach" [49] is at the foundation of SNAI; the strategy has the scope to promote IAs development, bridging Italian regional gaps [50].

In this framework, this paper provides new evidence on the relationship among per capita incomes—considered as a proxy of well-being—, local expenditures and territorial economic structure in Italian IAs.

The methodology employed is a panel data multilevel regression model [51], which fits well to the data at our disposal. The model cannot incorporate a spatial autoregressive parameter, since the area is not entirely made up of neighbouring municipalities, making it difficult to build a spatial contiguity matrix; we decided not to consider a more general connectivity matrix based on distances or other forms of spatial interrelations, since this choice would exclude many municipalities, giving a contribution to spatial correlation.

The paper contributes in two ways to the literature on regional development. Firstly, the territorial subdivisions employed (municipalities) are at the highest level of detail; to the best of our knowledge, this is the first time that such a study considers data following SNAI territorial subdivision. Secondly, there is an in-depth specification of local government expenditures, selected on the basis of the UN SDGs and targets.

The remainder of the study is organised as follows: Section 2 describes the dataset and the model employed; Section 3 highlights the results and discusses them; and Section 4 concludes the paper with some final comments.

## 2. Dataset and Methods

The area object of the present research is located in the territory belonging to three Italian regions, namely, Marche, Abruzzo, and Molise. They represent a portion of territory geographically located on the Middle Adriatic Sea (Figure 2). Abruzzo and Molise were previously a single region, until 1963, when Molise separated following a constitutional prescription. The chosen regions form a macro-region, the so-called "Marca Adriatica" [52]. Macro-regions are the core of a national debate on territorial reorganization of administrative functions, involving policymakers and scholars, such as economists, geographers, planners, and political scientists. "Marca Adriatica" will be referred to in this study as "MAM".

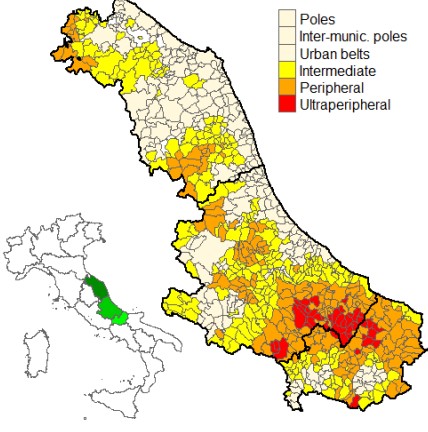

**Figure 2.** SNAI classification of the study area (IAs municipalities in yellow-to-red colours). The small figure highlights MAM regions (from top to bottom, Marche, Abruzzo and Molise).

The MAM macro-region consists of 665 municipalities, 440 of which are IAs municipalities (66.2%) (Table 1). Intermediate and peripheral municipalities represent the majority

of them (90.9% together). MAM is diversified with respect to SNAI classification, with Marche having only 45.1% of IAs municipalities, while in Abruzzo and Molise, this percentage amounts to 75.4% and 80.1%, respectively. Moreover, there are no ultra-peripheral municipalities in Marche.

**Table 1.** Distribution of municipalities into SNAI classes and macro-classes.

|  | Marche | | Abruzzo | | Molise | | Total | |
|---|---|---|---|---|---|---|---|---|
|  | n. | % | n. | % | n. | % | n. | % |
| Pole | 11 | 4.9 | 6 | 2.0 | 3 | 2.2 | 20 | 3.0 |
| Inter-municipal poles | 8 | 3.6 | 4 | 1.3 | 0 | 0.0 | 12 | 1.8 |
| Urban belts | 104 | 46.4 | 65 | 21.3 | 24 | 17.6 | 193 | 29.0 |
| Intermediate | 76 | 33.9 | 115 | 37.7 | 39 | 28.7 | 230 | 34.6 |
| Peripheral | 25 | 11.2 | 84 | 27.5 | 61 | 44.9 | 170 | 25.6 |
| Ultra-peripheral | 0 | 0.0 | 31 | 10.2 | 9 | 6.6 | 40 | 6.0 |
| Total Centres | 123 | 54.9 | 75 | 24.6 | 27 | 19.9 | 225 | 33.8 |
| Total IAs | 101 | 45.1 | 230 | 75.4 | 109 | 80.1 | 440 | 66.2 |
| Total Municipalities | 224 | 100 | 305 | 100 | 136 | 100 | 665 | 100 |
| % Interm/IAs | | 75.2 | | 50.0 | | 35.8 | | 52.3 |
| % Periph/IAs | | 24.8 | | 36.5 | | 56.0 | | 38.6 |
| % Ultra-Periph/IAs | | 0.0 | | 13.5 | | 8.3 | | 9.1 |

Table 2 presents some statistics relating to the last observational year (2016) with regards to territorial surfaces, population, and demographic density. IAs municipalities cover 62.7% of the total surface, with significant differences among regions, ranging from 42.9% of Marche to 83.4% of Molise. In terms of population, there are 874,116 people living in IAs (27.6% of the total). Molise is the least populated region, but conversely, it has the highest percentage of people living in IAs (60.5%). Demographic density in IAs is much smaller than in Centres (57 versus 247 inhabitants per square kilometre on the whole), with less variability with respect to Centres at the regional level.

**Table 2.** Summary statistics regarding SNAI macro-classes. Year 2016.

|  | Marche | Abruzzo | Molise | Total |
|---|---|---|---|---|
| Centres surfaces | 5228.0 | 3146.4 | 741.7 | 9116.1 |
| IAs surfaces | 3929.8 | 7682.4 | 3719.5 | 15,331.7 |
| Total surfaces | 9157.8 | 10,828.8 | 4461.2 | 24,447.8 |
| % IAs surf./Tot. surf. | 42.9 | 70.9 | 83.4 | 62.7 |
| Centres population | 1,287,151 | 845,448 | 122,478 | 2,255,077 |
| IAs population | 209,346 | 476,799 | 187,971 | 874,116 |
| Total population | 1,496,497 | 1,322,247 | 310,449 | 3,129,193 |
| % IAs pop./Tot. pop. | 14.0 | 36.1 | 60.5 | 27.9 |
| Demographic density—Centres | 246 | 269 | 165 | 247 |
| Demographic density—IAs | 53 | 62 | 51 | 57 |
| Demographic density—Total | 163 | 122 | 70 | 128 |

Table 3 illustrates mean per capita incomes in IAs municipalities in 2016, and the percentage variation between the first and the last observational periods.

Mean total incomes in 2016 tend to decrease from Poles to Ultra-peripheral areas. Between 2008 and 2016, the dynamics are negative in the three Centre classes and positive in the remaining IAs classes. This results in an income decrease in Centres (−1.22%) and in an increase in IAs (+1.09%). The Marche region shows higher per capita incomes in all of the classes; at the other end of the spectrum, Molise is the poorest region. The absolute values trend is decreasing in all three regions, while the rates of change differ among classes. The worst economic decline is registered in the Centres of Molise (−4.47% between 2008 and 2016). On the other hand, the biggest growth pertains to the IAs of Abruzzo (+2.61%).

**Table 3.** Mean per capita incomes (year 2016) and percentage change 2016/2008.

| | Marche | % | Abruzzo | % | Molise | % | Total | % |
|---|---|---|---|---|---|---|---|---|
| Poles | 13,940.10 | −4.35 | 12,799.28 | 0.45 | 11,569.23 | −7.13 | 13,242.22 | −3.39 |
| Inter-municipal poles | 12,229.81 | −3.43 | 10,109.40 | −3.40 | - | - | 11,523.01 | −3.42 |
| Urban belts | 11,783.15 | −1.20 | 9980.87 | 1.16 | 9162.77 | −4.03 | 10,850.31 | −0.79 |
| Intermediate | 11,101.68 | 0.81 | 9889.21 | 2.03 | 8699.64 | −2.81 | 10,088.14 | 0.85 |
| Peripheral | 10,387.62 | −0.02 | 10,059.82 | 4.25 | 8659.73 | −1.78 | 9605.64 | 1.55 |
| Ultra-peripheral | - | - | 9288.59 | 0.21 | 8225.09 | 1.90 | 9049.30 | 0.55 |
| Total Centres | 12,005.10 | −1.68 | 10,213.20 | 0.84 | 9430.16 | −4.47 | 11,098.81 | −1.22 |
| Total IAs | 10,924.93 | 0.61 | 9870.57 | 2.61 | 8638.12 | −1.87 | 9807.28 | 1.09 |
| Total Municipalities | 11,518.06 | −0.72 | 9954.82 | 2.16 | 8795.37 | −2.44 | 10,244.26 | 0.23 |

Per capita income levels highlight the differences in the regional economic structures; following the classical North–South dichotomy, economic conditions worsen going from Marche to Abruzzo to Molise [53].

Focusing the attention on macro-classes income dynamics, Figure 3 highlights the trend followed at the regional level. The evolution is the same for all of the macro-classes, with one exception for the Centres of Molise, which, since 2011, have experienced a stronger deterioration in per capita incomes.

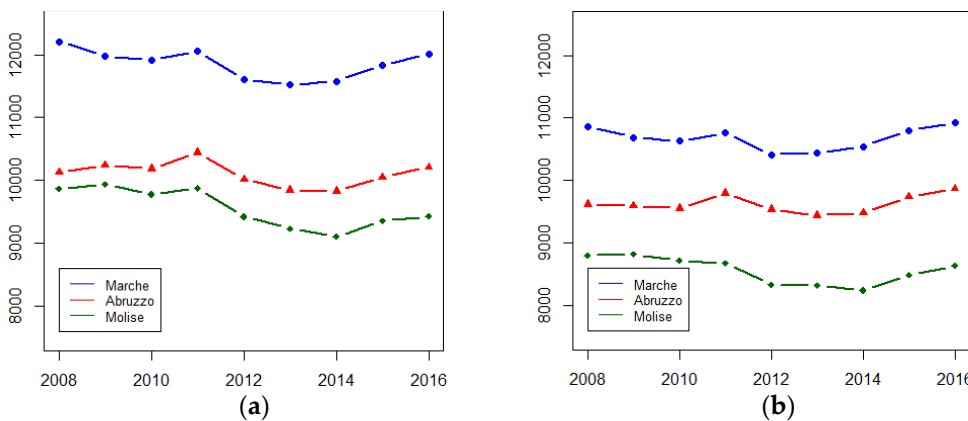

**Figure 3.** Mean per capita incomes dynamics in IAs municipalities (2008 constant prices). (**a**) Centres (**b**) Inner Areas.

In relation to the aims of the paper, the dataset employed consists of: (a) per capita incomes; (b) local administrations' expenditures; and (c) sectoral structure/employment data. Data (a)–(b) are at the municipal level, while data (c) are at the NUTS3 (province) level.

Regarding the territorial detail, municipalities have been chosen since SNAI explicitly acknowledges their role in IAs development [54]. Indeed, municipalities: "constitute the basic unit in the public policies decision process and, in the form of aggregations of neighbouring municipalities—inter-municipal local systems—they are privileged partners for the implementation of growth strategies" [55].

As to economic structure variables, they are not available at municipal level.

Per capita incomes data have been gathered from the Revenue Agency public database. Expenditure data have been collected from the AIDA/PA database of Bureau van Dijk. Local public administrations' financial statements include budgets, commitments, and payments; current and capital account payments have been considered in the present research. Province-level variables come from Italian National Statistical Institute (Istat) website.

The period under observation extends from 2008 to 2016. All of the monetary values have been converted to constant 2008 price values.

The single expenditures have been selected considering the guidelines set in the UN SDGs (United Nations, 2015; see Table 4). As a consequence, the following expenditure categories were chosen: (1) Education (EDU); (2) Cultural heritage (HER); (3) Youth and Leisure (Y&L); (4) Tourism (TOU); (5) Environment protection and planning (ENV); (6) Social policies (SOC); and (7) Economic development (ECO). The basic assumption underlying this choice is that these categories may have effects on well-being, as measured by per capita incomes.

**Table 4.** Expenditure items and categories, and related UN SDGs and targets.

| ITEMS | CATEGORIES | GOALS/TARGETS |
|---|---|---|
| • Preschool<br>• Primary schools<br>• Middle schools<br>• High schools | Education | ✓ Goal 4: Quality Education<br>*Targets 4.1, 4.2, 4.a* |
| • Libraries<br>• Museums<br>• Galleries<br>• Theatres | Cultural heritage | ✓ Goal 11: Sustainable cities<br>*Target 11.4* |
| • Youth policies<br>• Leisure facilities | Youth and Leisure | ✓ Goal 8: Decent work and economic growth<br>*Target 8.b*<br>✓ Goal 11: Sustainable cities<br>*Target 11.7* |
| • Tourism marketing<br>• Tourism services | Tourism | ✓ Goal 8: Decent work and economic growth<br>*Target 8.9*<br>✓ Goal 12: Responsible consumption and production<br>*Target 12.b* |
| • Urban planning<br>• Land care<br>• Waste and water management<br>• Protected areas<br>• Villages' sustainable growth | Environment protection and planning | ✓ Goal 6: Clean water and sanitation<br>✓ Goal 11: Sustainable cities<br>*Targets 11.6, 11.a*<br>✓ Goal 12: Responsible consumption and production<br>*Target 12.2*<br>✓ Goal 15: Life on land<br>*Targets 15.1, 15.2, 15.4, 15.5* |
| • Social housing<br>• Childhood and minors<br>• Disabled people<br>• Elders and households<br>• Disadvantaged persons<br>• Social security services | Social policies | ✓ Goal 1: No poverty<br>*Targets 1.2, 1.3, 1.5*<br>✓ Goal 3: Good health and well-being<br>*Target 3.5*<br>✓ Goal 11: Sustainable cities<br>*Target 11.1* |
| • Agriculture<br>• Industry<br>• Commerce<br>• Handicraft | Economic development | ✓ Goal 2: Zero hunger<br>*Target 2.4*<br>✓ Goal 8: Decent work and economic growth<br>*Target 8.3*<br>✓ Goal 12: Responsible consumption and production<br>*Target 12.7* |

As to NUTS3-level variables, the percent composition of total employment in the economic sectors (agriculture, industry, and services) has been included, together with the unemployment rate and the difference between exported and imported goods by road transport. This latter variable has been considered as a proxy of total net exports. It is important to underline that the variables employed are the only ones available at the NUTS3 subdivision.

As mentioned in the Introduction, the model suitable for the study is a multilevel panel data regression [56]; in this case, the model will be non-nested, since the interest is in analysing both territorial and time behaviour of the municipalities. The second-level variables are provinces and time.

Starting from the general matrix formulation, the model is:

$$y = \mathbf{X}\boldsymbol{\beta} + i_T \otimes \left[ \mathrm{bdiag}\left(i_{M_j}\right) \times \boldsymbol{\alpha} \right] + \boldsymbol{\delta} \otimes i_N + \boldsymbol{\xi} \tag{1}$$

where:

- $y_t = \left[ y_{11t}\, y_{21t} \cdots y_{M_1 1t}\, y_{12t}\, y_{22t} \cdots y_{M_2 2t} \cdots y_{1Jt}\, y_{2Jt} \cdots y_{M_J Jt} \right]'$, with $J$ being the number of provinces, is the vector of the $\sum_{j=1}^{J} M_j = N$ observations at time $t$;
- $y = [y'_1\, y'_2 \cdots y'_t \cdots y'_T]'$ is the $NT$-dimensional vector of all the observations in the panel, obtained by stacking the $T$ vectors (one each observation time) $y_t$;
- $\mathbf{X} = [\mathbf{X}_1\,;\, \mathbf{X}_2\,;\, \ldots;\, \mathbf{X}_t\,;\, \ldots;\, \mathbf{X}_T]$ is the $(NT \times K)$ matrix of the independent variables (possibly second-level variables at the province level, if they are considered only in the fixed effect part of the model);
- $\boldsymbol{\beta}$ is the $K$-dimensional vector of fixed effects parameters;
- $i_h$ is the $h$-dimensional vector of ones;
- bdiag (.) denotes a block-diagonal matrix;
- $\boldsymbol{\alpha} = \left[ \alpha_1\, \alpha_2 \ldots a_j \ldots \alpha_J \right]'$ is the vector of the random effects relating to the second-level aggregation into provinces;
- $\boldsymbol{\delta} = [\delta_1\, \delta_2 \ldots \delta_t \ldots \delta_T]'$ is the vector of the random effects relating to the second-level aggregation in time;
- $\boldsymbol{\xi} = [\xi'_1\, \xi'_2 \ldots \xi'_t \ldots \xi'_T]'$ is the $NT$-vector of residual random noise.

Equation (1) can be reformulated, with respect to the single observation (i.e., a municipality), as follows:

$$y_{ijt} = x'_i \boldsymbol{\beta} + \alpha_j + \delta_t + \xi_{ijt} \tag{2}$$

where $x'_i$ is the $p$-dimensional vector of the independent variables.

The multilevel model employed in this paper assumes the final form:

$$\begin{aligned} \log Y_{ijt} = {}& \beta_1 \log EDU_{ijt} + \beta_2 \log HER_{ijt} + \beta_3 \log Y\&L_{ijt} + \beta_4 \log TOU_{ijt} + \beta_5 \log ENV_{ijt} \\ & + \beta_6 \log SOC_{ijt} + \beta_7 \log ECO_{ijt} + \beta_8 \log AGR_{jt} + \beta_9 \log IND_{jt} \\ & + \beta_{10} \log SER_{jt} + \beta_{11} \log UNE_{jt} + \beta_{12} \log TRA_{jt} + \alpha_j + \delta_t + \xi_{ijt} \end{aligned}$$

where the meanings of each of the symbols have already been presented. Log-transformations of the variables have been taken in order to achieve the typical goals of analyses employing economic variables such as income, that is, approximate normality and stabilization of the variance [57]. Owing to the log $Y$–log $X$ transformation, the estimated parameters assuming small values will be interpreted as the (approximated) percent variation of the dependent variable according to a unit variation of the independent features.

## 3. Results and Discussion

As an initial step, three figures illustrate the main features of the variables. Figure 4 shows the territorial distribution of per capita incomes in the observation period in IAs. The municipalities belonging to the last class (€12,000–16,000 per capita) go from a minimum of 14 in year 2013, to 31 in 2016. At the opposite extreme, the poorest municipalities

(€6000–8000) range from a maximum value of 71 in 2014, to a minimum of 41 in 2009. The most represented province in the last class is L'Aquila (in Abruzzo), with 6 municipalities in 2013 and 13 in 2016. On the other side, Campobasso (in Molise) has 41 municipalities in the lowest class in 2014, and 24 in 2009.

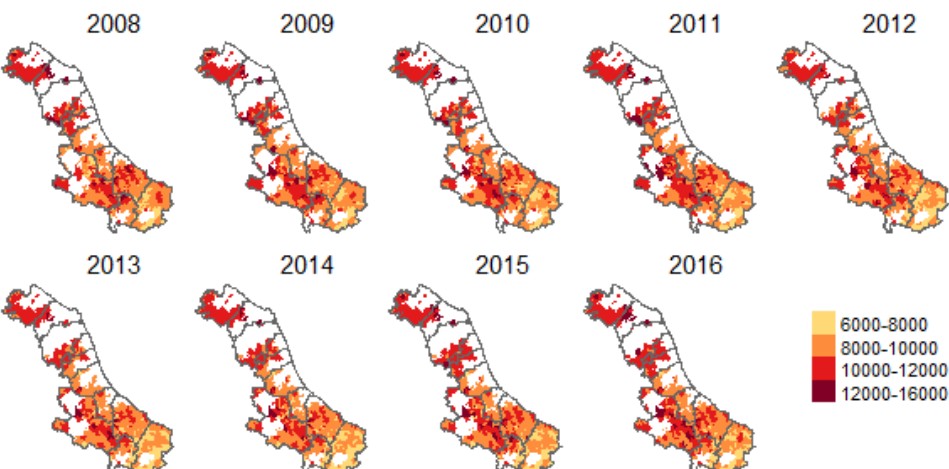

**Figure 4.** Per capita incomes in MAM inner areas municipalities. Years 2008–2016.

Figure 5 reports the total expenditures at the municipal level, in order to give an idea of the absolute level of per capita expenditures. As can be observed, total expenditures present high variability, measured in terms of the coefficient of variation, which ranges from a minimum of 88% in 2012 to a maximum value of 162% in 2015.

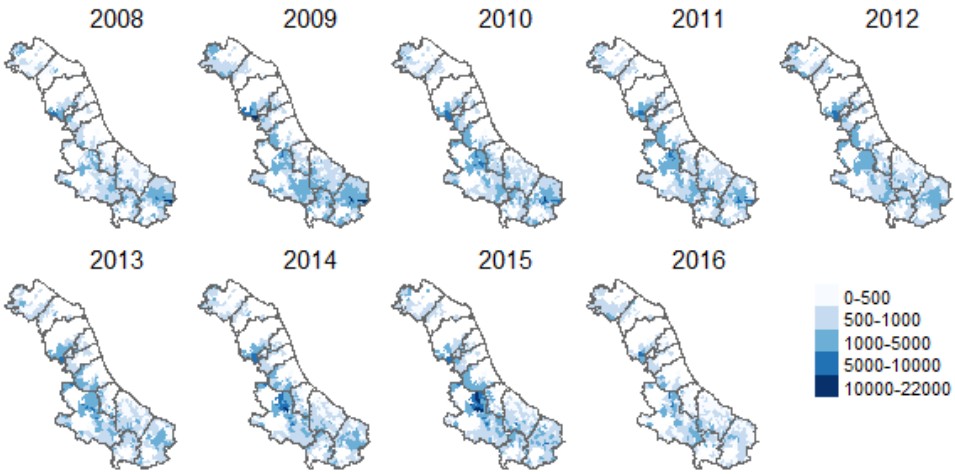

**Figure 5.** Mean total expenditures of municipal administrations. Years 2008–2016.

Figure 6 presents the dynamics of independent variables. In particular, Figure 6a reports mean per capita expenditures by category. As can be seen, the most important category is that of Environment protection and planning expenditures. The amount of mean total expenditures is procyclical, meaning that local administrators react slowly to economic changes. This behaviour is similar to central administrations' operating mode [44]. In this regard, Kolluri and Wahab [58] found evidence that in developed countries, government budgets are not responsive to economic cycles. As to 2016, the sharp decline could be due to the reduction of regional fund transfers towards municipalities, particularly for the *ENV* category. The decisions of local administrators about budgets are conditioned by the level of regional expenditure [59].

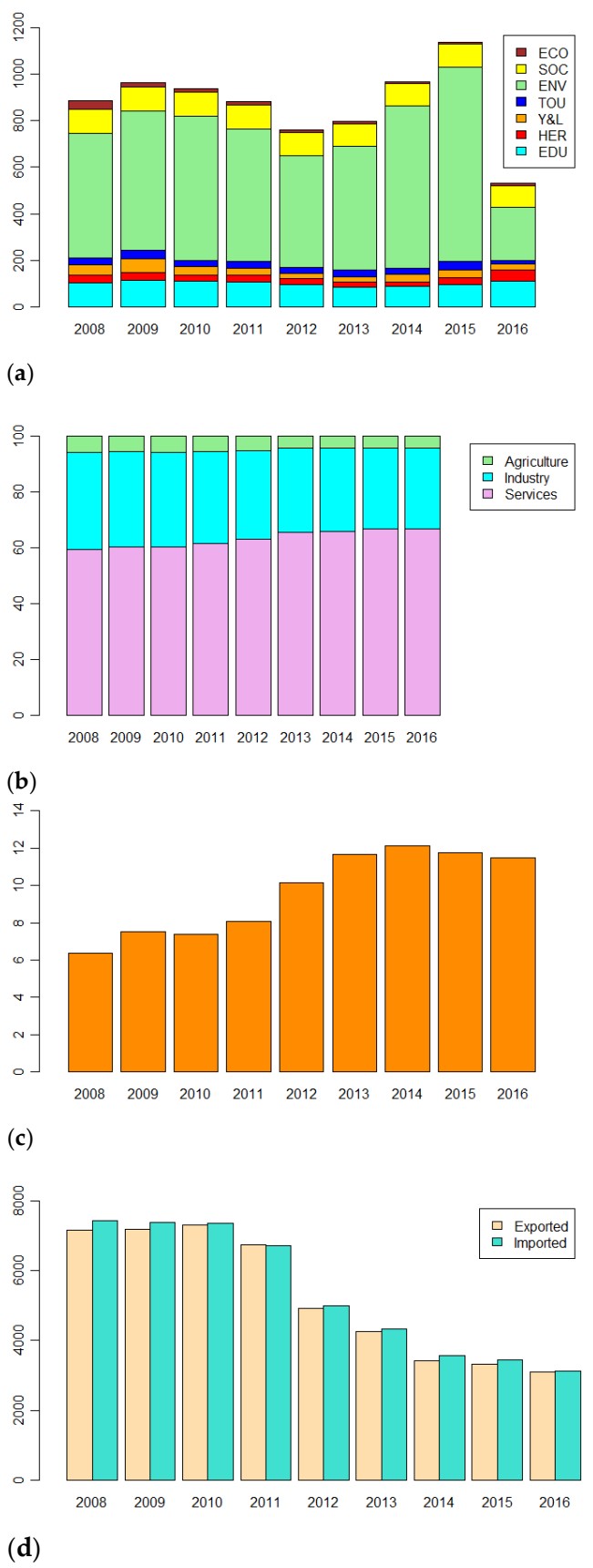

**Figure 6.** Independent variables dynamics in the MAM study area. Years 2008–2016. (**a**) Mean per capita expenditures by category. (**b**) Average percentage employment structure. (**c**) Unemployment rate. (**d**) Exported and imported goods by road transport (thousands of tons).

Figure 6b shows the average percentage of employment structure. The distribution of employees among productive sectors reflects the economic structure of the study area. It is evident that there is a slow but constant growth of the percentage of employment in the services sector, with a simultaneous small decline in industry and agriculture.

Regarding the unemployment rate, Figure 6c initially shows a strong increase during the crisis years of 2012–2014, and then a slower decrease starting from 2015, the first year of economic recovery.

Finally, Figure 6d illustrates the dynamics of extra-provincial exported and imported goods by road transport. The difference represents net exports by road transport, regarded as a proxy of net trade results. Both the quantities show a decline, especially in 2012, without presenting the capacity of returning to pre-crisis conditions after 2014.

Table 5 shows results from parametric estimation of model (3). The 95% confidence intervals have been obtained by means of a resampling procedure on 1000 bootstrap replicates. As can be observed, four of the fixed-effects parameters are not significantly different from zero: *SOC*, *AGR*, *SERV*, and *TRA*. In particular, although regional spending on social protection is a fundamental component of individual well-being [60], local governments' financial commitment in social policies can result as "indifferent" to the real social needs [61].

**Table 5.** Estimation results for model (3). The grey backgrounds highlight non−significant parametric values.

| Random effects: | | | | | |
|---|---|---|---|---|---|
| *Groups* | *Name* | *Variance* | *Std.Dev.* | *95% CI* | |
| Prov | Intercept | 0.0058 | 0.0761 | 0.0393 | 0.1085 |
| Time | Intercept | 0.0003 | 0.0185 | 0.0068 | 0.0272 |
| Residual | | 0.0109 | 0.1042 | 0.1017 | 0.1063 |
| **Fixed effects:** | | | | | |
| *Variable* | *Estimate* | *Std.error* | *t-value* | *95% CI* | |
| logEDU | −0.0165 | 0.0018 | −9.109 | −0.0198 | −0.0128 |
| logHER | 0.0098 | 0.0012 | 8.459 | 0.0077 | 0.0122 |
| logY&L | 0.0055 | 0.0013 | 4.277 | 0.0029 | 0.0079 |
| logTOU | 0.0105 | 0.0010 | 10.105 | 0.0084 | 0.0125 |
| logENV | 0.0138 | 0.0018 | 7.483 | 0.0102 | 0.0175 |
| logSOC | 0.0035 | 0.0018 | 1.940 | −0.0001 | 0.0071 |
| logECO | 0.0033 | 0.0012 | 2.771 | 0.0008 | 0.0057 |
| logAGR | −0.0219 | 0.0228 | −0.964 | −0.0658 | 0.0239 |
| logIND | 0.1672 | 0.0809 | 2.068 | 0.0079 | 0.3281 |
| logSERV | 0.2117 | 0.1554 | 1.362 | −0.0938 | 0.5138 |
| logUNE | −0.0338 | 0.0125 | −2.706 | −0.0594 | −0.0086 |
| logTRA | −0.0041 | 0.0074 | −0.557 | −0.0192 | 0.0092 |

Among the significant first-level variables, *EDU* is the only one negatively related to per capita incomes. This category primarily includes expenditures related to public school buildings—in particular, for their maintenance. Some authors have found a positive relationship between education spending and economic growth [23]. In general, education has a positive effect on human resources, by means of the creation of skilful, trained workers. This is particularly true when dealing about a higher educational level [21]. On the other side, IAs municipalities spend only for low educational level, because they have almost exclusively primary schools. This means that these expenditures do not succeed in having a revenue in terms of higher human capital. A negative association could depend also on the low efficiency of public expenditure for education [62].

All the other unit-level parameters have a positive sign; in particular, *ENV* and *TOU* expenditures present the highest values, equal to 0.0138 and 0.0105, respectively; this means an approximated percent variation of per capita incomes in the order of around 1% when a percent unitary increment of these variables is postulated. *HER* reaches a value only slightly lower. *ENV* comprises some voices which are very important for the issue of development, especially those referring to protected areas and to villages' sustainable growth. Moreover, a positive correlation has been found [63] between eco-efficiency and happiness, linking countries' general well-being to environmental conditions. Expenditures for protected areas promote an integrated endogenous development model, based on the enhancement of local natural resources from the perspective of a place-based approach [64]. Protected areas offer important recreational opportunities [65], and visitors appreciate their natural ecosystems and high levels of biodiversity [66]. National parks, in particular, attract many visitors [67], with positive effects on local economies [68]. Spending for villages' sustainable growth is part of the wider strategy regarding Smart villages, an EU project for the revitalization of rural areas [69].

Tourism public expenditures are a driving force for local tourism development [70]; indeed, tourism could be a substantial economic growth factor [71,72]. In IAs, it is based on the exploitation of nature, and this feature differentiates it from traditional tourism, thus requiring ad hoc policies [73]. Consequently, it would be necessary to pay special attention to amenities resources [74,75] and territorial identities [76]. The development of new forms of tourism in IAs could favour a "proactive conservation of landscape" [77]. Cultural heritage expenditures may constitute another fundamental factor of resilience for IAs. The enhancement of cultural resources favours youth employment and the creation of income opportunities for local residents [78]. Moreover, the combined effect of cultural heritage and cultural tourism may play an important role in regional economic development [79]. Finally, culture may also increase the levels of social capital and trust among members of a society [80]. The remaining expenditure categories, *Y&L* and *ECO*, though being statistically significant, have a minor effect on per capita incomes. In particular, expenditures for economic development may have positive effects—on one side, on corporate profitability by increasing firm productivity, and on the other side, by helping to lower firms' input costs, e.g., through public infrastructure provision [81]. Furthermore, local governments might attract new firms by offering state aid (e.g., subsidies) in order to create long-term jobs [82].

Among province-level variables, only two have been found to have a significant effect on incomes.

The rate of employment in the industrial sector in the IAs of the study region is much higher than the mean national value [83]. This means that the secondary sector has an important role in the economies of IAs municipalities. The so-called *tipping point* [84], that is, the GDP level at which the employment in the industrial sector starts to decrease, might not yet be reached in these areas, and the employment rate in the industry might still be linked in a direct way to per capita incomes.

Lastly, a negative relationship exists between incomes and unemployment. This is a confirmation, at local level, of the very well−known Okun's law, linking per capita GDP growth with a reduction in unemployment. The variations of employment are an important indicator of the social impact of the economic crisis [85]. Indeed, it has been found [86] that employment tends to return to pre-crisis levels at a slower rate than output, amplifying the effects on workers and their households.

As to random effects (Table 6), they confirm the impressions emerging from the former description of the database: per capita incomes are higher in those provinces presenting better economic and social features [87].

**Table 6.** Marginal random effects for model (3).

| Province | | Year | |
| --- | --- | --- | --- |
| Pesaro e Urbino | 0.0651 | 2008 | −0.0050 |
| Ancona | 0.1721 | 2009 | −0.0056 |
| Macerata | 0.0411 | 2010 | −0.0042 |
| Ascoli Piceno | 0.0628 | 2011 | 0.0137 |
| Fermo | 0.0336 | 2012 | −0.0129 |
| L'Aquila | 0.0720 | 2013 | −0.0149 |
| Teramo | −0.0458 | 2014 | −0.0122 |
| Pescara | −0.0400 | 2015 | 0.0124 |
| Chieti | −0.0245 | 2016 | 0.0403 |
| Campobasso | −0.1085 | | |
| Isernia | −0.0307 | | |

The effect of time, and therefore of economic crisis, is very well represented in Table 6, which shows negative values for all the years from 2008 to 2014, with the only exception of 2011, which registers a small positive value, somewhat surprisingly—very different from the national GDP trend (Istat, 2018). Moreover, since the global time effect is very small with respect to the provincial effect (Table 6), the combined random effects (province + time) reflect the local structural economic and social starting conditions (Figure 7); this means that, during economic crises, even the best local expenditure policies need some time to tackle the negative effects on income.

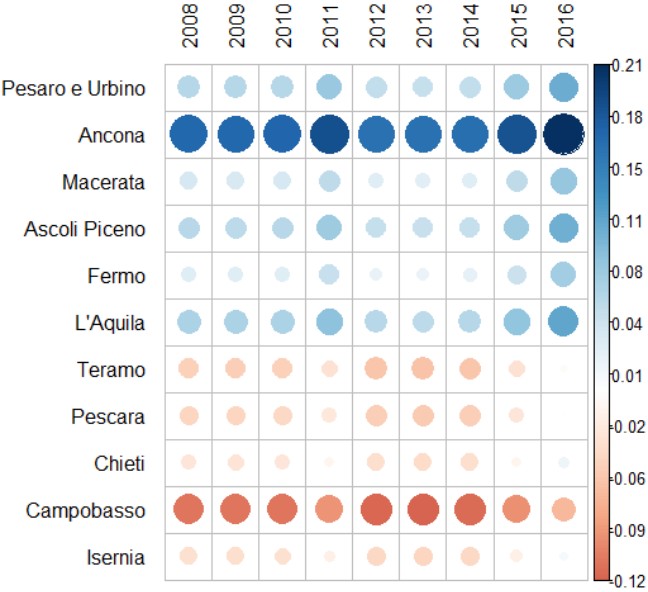

**Figure 7.** Combined random effects $Prov_j + Time_t$ for model (3).

## 4. Conclusions

This paper has shed some light on per capita incomes—considered as a proxy of well-being—in the municipalities belonging to the Middle Adriatic Sea MAM macro-region. In particular, it has provided empirical evidence about income determinants such as local public expenditures and economic structure.

As to methodological choices, it has been decided not to implement—within the panel multilevel model employed—the joint effect $(Prov \times Time)_{jt}$ for two reasons: first, the time effect is statistically significant, but not as strong as could be expected; and second, considering the joint effect would imply the estimation of too large a number (99) of parameters.

The results highlighted the importance of some categories of productive public expenditures, namely, those concerning Environment protection and planning, Tourism and Cultural heritage. These categories are related to the UN Goals for "Sustainable cities", "Life on land", "Responsible consumption and production", and "Economic growth". By contrast, expenditures relating to Social policies, linked to the Goals "No poverty" and "Good-health and well-being", were revealed to be not statistically significant. In contrast to what has been claimed in other studies (not performed at the municipal level), a significant but negative relationship has been found between Education expenditures and incomes. These findings call for some considerations about both the effectiveness of local public policies and the re-allocation of global expenditures among their various components. Municipalities could also try to raise funds to assign to productive expenditures through the recourse to public–private partnerships, and/or financial instruments for sustainable development projects (e.g., social impact bonds and green bonds). Since IAs municipalities are characterized by small dimensions, a new approach is needed in order to exploit scale dimensions to find new resources on financial capital markets. Another issue calls into question the promptness of public interventions, which should contrast income declines during recession phases (anti-cyclic interventions). Indeed, public spending in the short run supports income levels, but its effects in the long run have yet to be verified.

As to structural variables (second-level fixed effects), only two of them have proved to be statistically significant, that is, the share of employees in the industrial sector and the unemployment rate—this latter in a negative way.

The random effects estimated in the panel model highlight the importance of economic features at the provincial territorial scale in determining income levels. The time effect, as could be expected, does not play an important role, since the years under observation cover an economic stagnation phase.

The relevance of industry suggests that it could be useful to increase expenditures in this sector, considering also its interactions with the other sectors—agriculture and services—and the probable multiplicative beneficial effects it would bring to the whole economy. Furthermore, the presence of significant food-and-wine resources in MAM inner areas indicates that the implementation of "Food districts" might be helpful for income growth. Food districts, established by Italian Law 205/2017, are a new instrument aiming to promote local development, cohesion, and social inclusion, through the integration of activities characterized by territorial proximity, in a place-based approach. At the same time, the presence of environmental and cultural assets suggests to strengthen policy choices in the tourism sector, also taking into account its relationships with local amenities. Consequently, it is advisable to implement integrated policies oriented both to tourism and to the protection and enhancement of territorial resources.

Further insights, as well as topics for new studies, will be obtained by considering the effects of both public expenditures and the local economic structure on income, in light of the recent issues raised by pandemic waves.

**Author Contributions:** Conceptualization, L.R. and L.M.; methodology, L.R. and L.M.; formal analysis, L.R.; data curation, P.D.R.; writing—original draft preparation, L.R, P.D.R. and L.M.; writing—review and editing, L.R., P.D.R. and L.M. All authors have read and agreed to the published version of the manuscript.

**Funding:** This research received no external funding.

**Institutional Review Board Statement:** Not applicable.

**Informed Consent Statement:** Not appicable.

**Data Availability Statement:** Data will be made available on request.

**Conflicts of Interest:** The authors declare no conflict of interest.

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
