# Peer review of "Modelling Income Drivers in Peripheral Municipalities: The Case of Italian Inner Areas"

_sustainability, doi:10.3390/su142214754_

Round 1

Reviewer 1 Report

(1)Why did you choose 2008-2016? of course  I know 2008 is that the years following the outbreak of the global crisis, why select 2016 as the end year?

(2)Why did you choose  Marche, Abruzzo and Molise? How to reflect the representativeness of the three regions? Please add explanation.

(3) what did you select  Environment protection and planning, Tourism, and Cultural heritage as  the impact factors of public expenditure categories?

(4) P347-350, about which shows negative values for all the years from 2008 to 2014, with the only exception of 2011, which registers a small positive value”, please explain the reason about the registers a positive value in 2011.

(5)P 351-353 The combined random effects (province + time) reflect the structural economic and social starting conditions, since the global time effect is very small with respect to the provincial effect”, please describe the results in detail and explain the reason.

Author Response

(1)Why did you choose 2008-2016? of course  I know 2008 is that the years following the outbreak of the global crisis, why select 2016 as the end year?

Thank you for the observation; we changed lines 25-26 as follows: “the years going from the outbreak of the global financial crisis and the beginning of the economic recovery in Italy”.

(2)Why did you choose Marche, Abruzzo and Molise? How to reflect the representativeness of the three regions? Please add explanation.

We chose the three regions because “The chosen regions form a macro-region, the so-called “Marca Adriatica”. Macro-regions are the core of a national debate on territorial reorganization of administrative functions, involving policymakers and scholars, such as economists, geographers, planners, and political scientists.” (lines 110-113).

Regarding the representativeness of our area, we can state that the municipalities in MAM inner areas are 440, that is, more than 10% of Italian IAs municipalities (which are 4,185); moreover, the regions belong to Central (geographic) Italy, the national area which tends to have mean values for economic entities (“natural” good estimator of unknown economic parameters)

(3) what did you select “Environment protection and planning”, “Tourism”, and “Cultural heritage” as the impact factors of public expenditure categories?

Those three sectors of public expenditure have been commented on the basis of the results of regression model (Table 5): among the first-level (municipal) variables, they are significant from a statistical point of view, and show the highest positive estimated values (positive effects on per capita incomes).

(4) P347-350, about “which shows negative values for all the years from 2008 to 2014, with the only exception of 2011, which registers a small positive value”, please explain the reason about the registers a positive value in 2011.

Thank you for the notation. Indeed, it is very difficult to make realistic hypotheses on the causes of a one-year different behaviour; a possible, coherent conjecture refers to the fact that 2011 was a Population Census year in Italy. It is very frequent a reduction of the population as a result of a Census, especially in small municipalities (which, normally, tend to hide a reduction in resident population): as a consequence, the denominator in the ratio GDP/population reduces, thus increasing the value of the ratio. In the years following the Census one, the municipal authorities slowly tend to (slowly) reintegrate the population, until returning to precensal level.

(5)P 351-353 “The combined random effects (province + time) reflect the structural economic and social starting conditions, since the global time effect is very small with respect to the provincial effect”, please describe the results in detail and explain the reason.

Thank you very much for the observation; we have better explained our thoughts by rewriting lines 350-354 as follows: “Moreover, since the global time effect is very small with respect to the provincial effect (Table (6)), the combined random effects (province + time) reflect the local structural economic and social starting conditions (Figure 7); this means that, during economic crises, even the best local expenditure policies need some time to tackle the negative effects on income.”

Reviewer 2 Report

I find the paper to be an interesting read. It wonder be interesting to learn some charateristics of the population (such as ages, marital status, etc.) in the three areas. This is because different populations might value the public expenditure such as education differently. 

Author Response

We are grateful for the suggestion, but for the years considered (with the exception of the Census year 2011) there are no such data at municipal level.

Reviewer 3 Report

The authors explored the relationship among per capita incomes, local expenditures and territorial economic structure in Italian inner areas. Its a secondary data study and used panel multilevel regression model. The methodology is sound and well implemented. The results are presented in attractive way and easy to understand. 

Language is fine however, minor spell check is required. 

Author Response

Thank you so much for your very welcome compliments!